

# Introduction to the project VAHINE: VAriability of vertical and tropHIc transfer of diazotroph derived N in the south wEst Pacific

**S. Bonnet[1,2], T. Moutin[1], M. Rodier[3], J.M. Grisoni[4], F. Louis[4,5], E. Folcher[6], B. Bourgeois[6], J.M. Boré[6], A. Renaud[6]**

[1] {Aix Marseille Université, CNRS/INSU, Université de Toulon, IRD, Mediterranean Institute of Oceanography (MIO) UM 110, 13288, Marseille, France}

[2] {Institut de Recherche pour le Développement, AMU/ CNRS/INSU, Université de Toulon, Mediterranean Institute of Oceanography (MIO) UM 110, 98848, Noumea, New Caledonia}

[3] {Institut de Recherche pour le Développement, Université de la Polynésie française - Institut Malardé - Ifremer, UMR 241 Ecosystèmes Insulaires Océaniens (EIO), IRD Tahiti, PB 529, 98713 Papeete, Tahiti, French Polynesia}

[4] {Observatoire Océanologique de Villefranche-sur-Mer, UMS 829, Villefranche-sur-Mer, France}

[5] {Centre National de la Recherche Scientifique, UMR 7093, Observatoire Océanologique de Villefranche-sur-Mer, Laboratoire d'Océanographie de Villefranche-sur-Mer, Villefranche-sur-Mer, France}

[6] {Institut de Recherche pour le Développement, 98848, Noumea, New Caledonia}

Correspondence to: S. Bonnet (sophie.bonnet@ird.fr)



**Abstract**
At the global scale, $N_2$ fixations provides the major external source of reactive nitrogen to the
surface ocean, before atmospheric and riverine inputs, and sustains ~50 % of new primary
production in oligotrophic environments. The main goal of the VAHINE project was to study
the fate of nitrogen newly fixed by diazotrophs (or diazotroph-derived nitrogen) in oceanic
food webs, how it impact heterotrophic bacteria, phytoplankton and zooplankton dynamics,
stocks and fluxes of biogenic elements and particle export. Three large-volume (~50 m3)
mesocosms were deployed in a tropical oligotrophic ecosystem (the New Caledonia lagoon,
south-eastern Pacific) and intentionally fertilized with ~0.8 µM of dissolved inorganic
phosphorus (DIP) to stimulate diazotrophy and follow subsequent ecosystem and fluxes
changes. VAHINE was a multidisciplinary project involving close collaborations between
biogeochemists, molecular ecologist, chemists, marine opticians and modelers. This
introductory paper describes in detail the scientific objectives of the project as well as the
implementation plan: the mesocosm description and deployment, the selection of the study
site (New Caledonian lagoon) and the logistical and sampling strategy. The description of the
main hydrological and biogeochemical conditions of the study site before the mesocosms
deployment and during the experiment itself is then detailed, and a general overview of the
papers published in this special issue is presented.











**1    General context and objectives of the VAHINE project**
Climate change is now widely recognized as the major environmental problem facing the
globe (IPCC, 2014) and is at the heart of human, environmental and economical issues. On a
global scale, the oceanic biological carbon pump (BCP) influences climate trends: it consists
of the photosynthetic fixation of carbon dioxide ($CO_2$) by oceanic algae (phytoplankton) in
the upper illuminated ocean, followed by the downward flux of some of this material mainly
due to gravitational settling. The BCP transfers approximately 5-15 GT of carbon (C) from
the surface ocean to the oceans interior every year (Henson et al., 2011).
The efficiency of our oceans to take up excess $CO_2$ largely depends on the availability of
fixed nitrogen (N) (Falkowski, 1997) in the surface ocean. In the vast nitrate ($NO_3^-$)-limited
oligotrophic gyres, which cover ~60 % of the global ocean surface, fixed N is principally
provided through the biological fixation of atmospheric dinitrogen ($N_2$) by $N_2$-fixing (or
diazotrophic) organisms (Karl et al., 2002). Diazotrophs fix $N_2$ gas dissolved in seawater (the
largest reservoir of N on Earth) into ammonium and organic N compounds. At the global
scale, they provide the major external source of N for the ocean, before atmospheric and
riverine inputs (Gruber, 2004), and act thus as 'natural fertilizers', contributing to sustain life
and the BCP through the so called '$N_2$-primed prokaryotic C pump' (Karl et al., 2003; Karl et
al., 2012).
Important progress on the magnitude and the ecological role of marine $N_2$ fixation in
biogeochemical cycles has been made by the international oceanographic community over the
last two decades. They include the landmark discovery of unicellular diazotrophic organisms
of pico- and nanoplanktonic size termed UCYN, e.g. (Zehr et al., 2001), and new and
unexpected ecological niches where diazotrophs are active, such as N-rich oxygen minimum
zones, e.g. (Dekaezemacker et al., 2013; Fernandez et al., 2011). Thus, we have gained a
much better understanding of this process. However, a critical question that remains poorly
studied is the fate of N newly fixed by diazotrophs (or diazotroph derived N, hereafter
referred to as DDN) in oceanic food webs, and its impact on $CO_2$ uptake and export (BCP)
(Mulholland, 2007). The VAHINE project proposes a scientific contribution to answer these
questions, based on a combination of experimentation and modelling involving recently
developed innovative techniques. The main scientific questions of the VAHINE project were:

i) What is the primary route of transfer of DDN through the planktonic food web, i.e. is DDN
preferably transferred to large size (e.g. diatoms), small size (pico-, nanophytoplankton)
phytoplankton, or to the microbial food web? How much DDN is transferred to zooplankton?



ii) Does the development of diazotrophs influence auto- and heterotrophic plankton diversity
and gene expression dynamics, as well as pico-, nano-, and microphytoplankton abundances?
Do they influence zooplankton dynamics?
iii) Does the development of diazotrophs significantly modify the stocks, fluxes, ratios of the
major biogenic elements (C, N, P)?
iv) Does the development of diazotrophs influences the efficiency of carbon export? Is this
export direct or indirect?
A detailed literature review on our knowledge regarding the fate of DDN in the ocean is
provided in the synthesis article of the present issue (Bonnet et al., In prep.). Here we will
focus on the technical challenges and the methods developed to answer the scientific
questions of the project.
Studying the fate of DDN in the ocean is technically complex. First, it requires appropriate
methodologies to trace the passage of DDN through the different components of planktonic
food web. During the VAHINE project, we intensively used high-resolution nanometer scale
secondary ion mass spectrometry (nanoSIMS) in combination with flow cytometry cell
sorting and $^{15}N_2$ labelling to trace the passage of $^{15}N$-labelled DDN into several groups of
non-diazotrophic phytoplankton and bacteria. This technique and results are extensively
presented in (Bonnet et al., Under Revision) and in the special issue (Berthelot et al.,
Submitted; Bonnet et al., Submitted) and will not be detailed here.
Second, it requires to monitor the chemical, biological and biogeochemical characteristics of a
water body affected by a diazotroph bloom for a long period of time (15-30 days) to be able to
track plankton community changes, track the N transfer in the different compartments of the
ecosystem (dissolved/particulate phases, small/large plankton, export material) and elaborate
biogeochemical budgets. Small-scale laboratory microcosm experiments have been frequently
used in ocean biogeochemical studies, but their limited realism can make extrapolations to
natural systems difficult to justify. They limit the duration of experiments to few days
(usually 24 to 72 h), the small volumes used (few liters maximum) limit the number of
parameters measured and they do not include the export terms. To overcome these
difficulties, we decided to use the technology of large-volume mesocosms. Mesocosms enable
to isolate water masses of several cubic meters from physical dispersion for several weeks,
without disturbing temperature and light conditions, taking into account the biological
complexity of the planktonic ecosystem at large scales, and thus provide a powerful approach



to maintain natural planktonic communities under close-to-natural self-sustaining conditions
for several weeks. Moreover, the responses obtained from mesocosms studies (isolated from
hydrodynamics) provide useful parameterizations for ecosystem and biogeochemical models.

## 5   2   Implementation of the VAHINE project

### 6   2.1   Mesocosms description and deployment

The mesocosms (surface 4.15 m$^2$, volume ~50 m$^3$, Fig. 1) chosen for this study are sea-going
mesocosms entirely transportable that can be used under low to moderate wind/wave
conditions (20-25 knots/2.5 wave height). They have been designed in the framework of the
DUNE project (Guieu et al., 2010; Guieu et al., 2014). They consist in large transparent bags
made of two 500 μm thick films of polyethylene (PE) and vinyl acetate (EVA, 19 %), with
nylon meshing in between to allow maximum resistance and light penetration (produced by
HAIKONENE KY, Finland) (Fig. 2). They are 2.3 m in diameter and 15 m in height and are
equipped with removable sediment traps for sinking material collection (Fig. 1, 2), which was
prerequisite to answer some of the questions of the project. In the framework of VAHINE, we
deployed three mesocosms (hereafter named M1, M2 and M3) to ensure a replication and
robustness of the data.
The mesocosms were made of three different parts (Fig. 1, 2): i) the main cylinder, rigidified
by five polyethylene rings maintaining the round shape of the bags and ending with two 8 cm
width PVC circles sandwiching the bags ii) the bottom cone (2.2 m height) also made of two
8 cm width PVC circles. It was equipped with the sediment trap system, on which is screwed
a 250 mL flask collecting sinking material, allowing an easy daily collection and replacement
by SCUBA divers, iii) the PE flotation frame supporting the bags and attached at three points
thanks to specific PVC cylindrical structures at the level of the upper ring and at the level of
the ring just below the sea-surface. The structure was equipped with six buoys insuring the
buoyancy of the system.
The mesocosms were moored using three screw anchors installed on the sea floor (25 m
depth). The three mesocosms were attached together and moored with the anchors screwed
120° from each other and connected to sub-surface buoys, which were themselves connected
to surface buoys. The complete setup was a solid mooring capable of absorbing the sea swell
while maintaining a supple and strong structure and ensuring that no tension was applied
directly to the bags. An in situ mooring line was installed on an independent screw anchor to
incubate subsamples collected from the mesocosms for production measurements (primary
production, N$_2$ fixation) and process studies under the same conditions as in the mesocosms.



A fifth independent screw anchor was installed to hold the two mobile plastic platforms
necessary to welcome the scientists and instrumentation for the daily sampling.
The mesocosms were deployed on January 13th 2013 (day 0) thanks to the assistance of four
professional SCUBA divers. The group of three main cylinders was first deployed and the
initial operations were performed on a coral shoal near the deployment site. The bags, cinched
by three small elastic ropes, were placed inside and fixed to the flotation frame at three places
using the designed PVC pieces. Once fixed, the system was transported to the deployment
site, and attached to the subsurface buoys located at the vertical of screw anchors. Small
ballasts were set up at the base of the bags and the elastic ropes released, allowing the main
cylinders to gently deploy vertically with the assistance of the SCUBA divers (Fig. 2e,f).
Once deployed, the main cylinders were left opened for 24 h to stabilize the water column
inside. The day after (day 1, January 14th), the divers closed the mesocosms by screwing
together the main cylinder and the bottom cone using eight nylon screws preventing any water
exchange between inside and outside the mesocosms (Guieu et al., 2010). During the entire
installation, the divers followed instructions to remain outside the bags to minimize
disturbance and potential contamination of the water column.

## 2.2   Selection of the study site

The mesocosms were deployed during austral summer conditions (January-February 2013) in
the oligotrophic New Caledonian coral lagoon (Noumea lagoon). New Caledonia is located in
the South West Pacific ocean, 1500 km east of Australia in the Coral Sea (Fig. 3a), and hosts
one of the three largest reef systems worldwide. It still displays intact ecosystems and its
ecological and patrimonial value has been recognized through its registration as a UNESCO
world heritage site. This site has been chosen for several reasons: i) it is a tropical low-
nutrient low-chlorophyll (LNLC) ecosystem strongly influenced by oceanic oligotrophic
waters inflowing from outside the lagoon (Ouillon et al., 2010). $NO_3^-$ and chlorophyll a (Chl
a) concentrations are typically $< 0.04$ µmol $L^{-1}$ and around 0.10-0.15 µg $L^{-1}$, respectively,
during the summer season (Fichez et al., 2010). ii) Primary productivity is N-limited
throughout the year (Torréton et al., 2010), giving $N_2$-fixing microorganisms a competitive
advantage over non-diazotrophic organisms. New Caledonian waters support high $N_2$ fixation
rates (151-703 µmol N $m^{-2}$ $d^{-1}$, Garcia et al., 2007), high *Trichodesmium* spp. abundances
(Dupouy et al., 2000; Rodier and Le Borgne, 2010, 2008) as well as unicellular diazotrophic
cyanobacteria (UCYN) (Biegala and Raimbault, 2008). The New Caledonian lagoon therefore





represented an ideal location to track the fate of DDN in the ecosystem and implement the
VAHINE project.
Before the VAHINE project, the mesocosms chosen for this study had only been deployed in
protected bays of the temperate Mediterranean Sea, which is not submitted to tide currents
and trade winds as New Caledonia is. In order to test the resistance of the mesocosms in a
tropical ecosystem submitted to trade winds (20-25 knots) and high tidal currents, and to
select the ideal location to deploy the mesoscosms inside the lagoon, we performed a pilot
study in March 2012 (i.e. one year before the VAHINE project). Four potential study sites
have been tested and the Tabou Reef (22°29.073 S - 166°26.905 E) located in close proximity
to Boulari passage (Fig. 3b, c) has been selected as the ideal location to implement the project
as it met the following specifications required for the technical deployment and sustainability
of the mesocosms: i) the site was protected from the dominant trade winds by the submerged
reef located less than one nautical mile from the study site, ii) it was located 28 km from the
New Caledonian coast at the exit of the lagoon and was strongly influenced by oceanic
waters, typical of a LNLC environment (see below, initial conditions), iii) it was 25 m-deep,
which is in the range required (17-25 m) to deploy 15 m high mesocosms and insure the
SCUBA divers security, iv) the seafloor was mainly composed of sand, which is a
prerequisite to implant to screw anchors in the substrate, v) it was low visited by amateur
yatchmen.

## 2.3   DIP fertilization

Dissolved inorganic phosphorus (DIP) availability has been reported to control $N_2$ fixation in
the southwest Pacific (Moutin et al., 2008; Moutin et al., 2005). To alleviate any potential DIP
limitation in the mesocosms and enhance a bloom of diazotrophs for the purpose of this study,
the mesocosms were intentionally fertilized with ~0.8 µmol $L^{-1}$ of DIP on the evening of day
4 (January 16[th]) of the experiment. We diluted 5.66 g of $KH_2PO_4$ in three 20-L carboys filled
with filtered surface seawater collected close to the mesocosms. The carboys were
homogenized and 20 L of each solution have then been carefully introduced in each
mesocosm from the bottom to the surface thanks a braided PVC tubing (inner diameter = 9.5
mm) connected to a Teflon pump (St-Gobain Performance Plastics) gradually lifted up during
the $KH_2PO_4$ fertilization to insure homogenization of the solution.
When deployed, the mesocosms naturally trapped different volumes of seawater and the
volume of each mesocosms had to be determined for biogeochemical budgets (Berthelot et
al., 2015). As DIP concentrations were measured at three selected depths (1 m, 6 m, 12 m)



before (evening of day 4) and after (morning of day 5) the fertilization, the delta DIP was
used to calculate the volume of each mesocosm based on the assumption that no DIP was
consumed during the night between day 4 and day 5. The DIP concentrations were
homogeneous over depth on day 5 and the mesocosm volumes were calculated as 52,790±490
L for M1, 42,620±430 L for M2 and 50,240±300 L for M3, with the uncertainties calculated
from standard deviation of triplicate DIP measurements.

## 2.4   Logistics, sampling strategy

As the mesocosms were moored 28 km off the coast, all the experimental work had to be
performed on site: scientific laboratories were setup on the R/V Alis (28.5 m) moored 0.5
nautical mile from the mesocosms, and on the Amédée sand island located one nautical mile
from the mesocosms (Fig. 3b, c), on which we set up a laboratory and accommodated
scientists for the duration of the VAHINE experiment.
Sampling in the mesocosms started on January 15$^{th}$ (day 2). It was performed daily for 23
days until February 6$^{th}$ at 7 am from the sampling platform moored next to the mesocosms.
Every day after collection, seawater samples were immediately carried out to the R/V Alis
and the Amédée for immediate processing.
Discrete samples were collected at three selected depths (1 m, 6 m, 12 m) in each mesocosm
and outside (hereafter termed 'lagoon waters') using a braided PVC tubing connected to the
Teflon PFA pump  activated by pressurized air from diving tanks, allowing to sample large
volumes with the least possible perturbation inside the mesocosms. For stocks measurements,
50-L PE carboys were filled at each depth of each mesocosm, immediately transported
onboard the R/V Alis for subsampling and samples treatments. For fluxes measurements
(primary production, bacterial production, $N_2$ fixation), samples were directly collected in
incubation bottles and transported onboard to skip the subsampling step and minimize the
time between collection, tracer spikes and incubation. For prokaryotic diversity and
expression measurements, 10-L carboys were filled (from M1 only) and carried out to the
Amédée laboratory for immediate processing. A total of 220 L were sampled every day from
each mesocosms, corresponding to ~10 % of the total mesocosms volume sampled at the end
of the 23-days experiment.
After seawater sampling, vertical CTD profiles were performed (around 10 am) using a SBE
19 plus Seabird CTD in each mesocosm and outside the mesocosms to obtain the vertical
structure of temperature, salinity and fluorescence. The CTD *in situ* fluorescence data were
fitted to the Chl *a* data from fluorometry measurements using a linear least squares regression.



Sediment traps were then collected daily from each mesocosm by two SCUBA divers (Fig.
2e, f1). They followed the same protocol everyday: they carefully hit the cone of the
mesocosms in case some sinking material was retained on the walls, waited for 15 minutes,
and collected the 250 mL flasks screwed to the trap system of each mesocosm and
immediately replaced it by a new one.
Vertical net hauls were performed every four days using a 30 cm diameter, 100 cm long, 80
μm mesh net fitted with a filtering cod end. On each sampling occasion, three vertical hauls
were collected from each mesocosm and lagoon waters, representing a total volume of 2.13
m$^3$, i.e. 4 % of the total mesocosm volume. This sampling strategy has been chosen to
minimize the effect of zooplankton catches on the plankton abundance and composition in the
mesocosms.
**2.5   Replicability among the mesocosms**
(Guieu et al., 2010; Guieu et al., 2014) have performed several mesocosm experiments in the
Mediterranean Sea, and demonstrated that the type of mesocosms used in the present study is
well adapted to conduct replicated process studies on the first levels of the pelagic food web
in LNLC environments. In order to evaluate the reproducibility among the three deployed
mesocosms during VAHINE, we calculated the coefficient of variation (CV, %) of the main
stocks and fluxes measured every day for 23 days for every sampling depth (Table 1, the
methods are described in detail in the publications composing this special issue). The CV
ranged from 4 to 42 % depending on the parameter considered. It was the lowest for TOC and
DON concentrations (4 and 9 %, respectively), which is very satisfying as these CV are close
to the precision of the methods themselves, indicating a good reproducibility between
mesocosms. It was the highest for $NO_3^-$ concentrations (42 %), which is consistent with the
fact that $NO_3^-$ concentrations were close to quantification limits of conventional methods
(~0.05 μmol L$^{-1}$) during the 23-days experiment: when the mean value is close to zero, the
CV approaches infinity and is therefore sensitive to small changes in the mean. For flux
measurements such as PP, BP and $N_2$ fixation, the CV was 29, 26 and 34%, respectively,
which is also satisfying given the natural spatial heterogeneity of plankton in the environment
due to aggregation, (Seebah et al., 2014), or to the buoyancy of some diazotrophs such as
*Trichodesmium* (Capone et al., 1997), which introduces some spatial, well known in the
natural environment for $N_2$ fixation (Bombar et al., 2015).
Another criterion to evaluate the consistency between mesocosms is to compare the evolution
of the biogeochemical conditions and the plankton community composition between




mesocosms. It is described in details in several articles of the present issue and only some
general features will be given here. As an example, bulk $N_2$ fixation rates averaged $18.5\pm1.1$
nmol N $L^{-1}$ $d^{-1}$ over the 23 days of the experiment in the three mesocosms (all depths averaged
together). The variance between the three mesocosms was low, $N_2$ fixation rates did not differ
significantly from the three mesocosms ($p<0.05$, Kruskall-Wallis test, (Berthelot et al., 2015)
and we consistently observed the same temporal dynamics over the three mesocosms, such as
the dramatic increase of rates from days 15 to 23 (they reached $27.3\pm1.0$ nmol N $L^{-1}$ $d^{-1}$). This
together indicates good replicability between the mesocosms (Bonnet et al., Submitted).
Molecular data also report a shift in the diazotrophic community composition around day 15,
with a bloom of UCYN-C consistently occurring in the three mesocsoms, see (Turk-Kubo et
al., 2015). The same feature was observed for *Synechococcus* abundances, which increased by
a factor of two since day 15 to day 23 in every mesocosm (Leblanc et al., this issue). Finally,
the diatom community which was very diverse during the first half of the experiment
suddenly shifted since ~day 10 and *Cylindrotheca closterium* consistently became the
dominant diatoms in the three mesocosms (Leblanc et al., Submitted). These observations,
together with the CV reported above indicate that the biogeochemical and biological
conditions were comparable between the three mesocosms.

## 19   3   Initial conditions and evolution of the core parameters during the
## 20     experiment

Initial hydrological and biogeochemical conditions (i.e. conditions in ambient waters the day
of mesocosms deployment - January 13[th], day 0) are summarized in Table 2. Seawater
temperature was 25.30°C, which is slightly lower than the classical temperature reported at
this season at the Amédée lighthouse (Le Borgne et al., 2010). Salinity was 35.15, a classical
salinity measured at this season at the Amédée lighthouse station (Le Borgne et al., 2010).
$NO_3^-$ and DIP concentrations were $0.04\pm0.01$ μmol $L^{-1}$ for both, and Chl *a* concentrations
from fluorescence data ($0.11$ μg $L^{-1}$) were typical of oligotrophic systems and are in the range
reported in the literature for this location (Fichez et al., 2010). Dissolved organic N (DON)
and P (DOP) concentrations were $4.65\pm0.46$ and $0.100\pm0.002$ and ambient $N_2$ fixation rates
$8.70\pm1.70$ nmol N $L^{-1}$ $day^{-1}$ before the mesocosms deployment.
Seawater temperature measured daily by vertical CTD profiles inside the mesocosms and in
the lagoon waters (Fig. 4a-d) gradually increased over the 23-days of the experiment from
25.50°C the day of the mesocosms closure (day 2) to 26.24°C on day 23. This warming is the
classical trend observed in New Caledonia along the summer season (Le Borgne et al., 2010).





The water column was not stratified over the course of the experiment, except the two first
days, which were characterized by a slight stratification inside and outside the mesocosms.
Data indicate therefore a good reproducibility between the three mesocosms and between the
mesocosms and the Noumea lagoon waters.
Salinity data (Fig. 4e-h) indicate a small and gradual increase in the three mesocosms during
the 23-days experiment (35.2 to 35.4) indicating a probable higher level of evaporation in the
mesocosms compared to the Noumea lagoon. Moreover, lagoon waters constantly receive
some low salinity waters from the coast due to rainfall advected by tide currents, which may
also explain the slightly lower salinity values measured in the Noumea lagoon (35.40)
compared to inside (35.47) at the end of the experiment.
$NO_3^-$ concentrations (Fig. 5a-d) remained below 0.1 µmol $L^{-1}$ during the whole experiment in
all mesocosms and in the lagoon waters. Average concentrations over the 23-days experiment
and the three depths samples were close to detection limits of the method (0.01 µmol $L^{-1}$) and
are thus difficult to quantify accurately: they were 0.04±0.02 µmol $L^{-1}$, 0.02 ±0.01 µmol $L^{-1}$,
0.02±0.02 µmol $L^{-1}$, and 0.06±0.04 µmol $L^{-1}$ in M1, M2, M3 and in the lagoon waters,
respectively. DIP concentrations (Fig. 5e-h) were also close to detection limits (0.005 µmol $L^-$
$^1$) and on average 0.04±0.01, 0.03±0.01 and 0.03±0.02 µmol $L^{-1}$ before the DIP fertilization
(days 2 to 4, hereafter called P0) in M1, M2 and M3 (average over the three depths). They
increased after the fertilization on day 5 to 0.73±0.07, 0.98±0.01, 0.77±0.03 µmol $L^{-1}$ in M1,
M2 and M3. The intensity of the DIP fertilization differed slightly among the mesocosms,
likely reflecting the different volume of the mesocosms (see above). Subsequently the DIP
concentrations decreased steadily towards initial concentrations by the end of the experiment:
0.03±0.01, 0.03±0.01 and 0.05±0.02 µmol $L^{-1}$ in M1, M2 and M3, respectively (average of
days 23 over the three depths). However, the DIP pool was first exhausted in M1 (day 14),
then M2 (day 19) and finally M3 (day 23). A more detailed description of the evolution of
stocks and fluxes of biogenic elements during the experiment can be found in (Berthelot et al.,

27   2015).

Chl *a* fluorescence was homogenous over the water column during the course of the
experiment (Fig. 4i-l). Chl *a* slightly increased (by 0.1 to 0.2 µg $L^{-1}$) in the three mesocosms
after the DIP fertilization on days 5 and 6. After day 6, they consistently decreased back to the
initial (before fertilization) concentrations of 0.12-0.15 µg $L^{-1}$. On days 12, 13 and 14, Chl *a*
concentrations re-increased dramatically to reach 0.61, 0.65 and 1.02 µg $L^{-1}$ in M1, M2 and
M3 at day 23, respectively, indicating that the three mesocosms were relatively synchronized
but the intensity of the phytoplankton bloom differed between the mesocosms, with a higher



increase observed in M3 compared to M2 and M1. In the lagoon waters, Chl *a* concentrations
also gradually increased over the experiment (concentrations reached 0.35 µg L$^{-1}$ at day 23)
but to a lower extend compared to that of the mesocosms.
**4    Special issue presentation**
The goal of this special issue is to present the knowledge gained regarding the fate of DDN in
a LNLC ecosystem based on the large dataset acquired during the VAHINE mesocosm
experiment. VAHINE was a multidisciplinary project involving close collaborations between
biogeochemists, molecular ecologist, chemists, marine opticians and modelers. Most of the
contributions to this special issue have benefited from this collective and collaborative effort.
The philosophies of the different papers composing the special issue are presented briefly
hereafter and a synthesis paper of all the multidisciplinary approaches used to answer the
main scientific questions of the VAHINE project is proposed at the end of the issue.
First, thanks to the high frequency (daily) sampling of the same water body for 23 days, this
project provided a unique opportunity to characterize the diversity of the planktonic
assemblage using several and complementary approaches, and investigate species successions
in relation to hydrological parameters, biogeochemical stocks and fluxes during a diazotroph
bloom in a LNLC ecosystem. By using PCR targeting a component of the nitrogenase gene
(*nifH*), sequencing and qPCR assays, (Turk-Kubo et al., 2015) fully characterized the
diazotroph community composition within the mesocosms and the New Caledonian
(Noumea) lagoon and calculated *in situ* growth and mortality rates for natural populations of
diazotrophs, which is rarely accomplished. This study provided the first growth rates for the
uncultivated UCYN-A2 and the UCYN-C phylotypes, and the first opportunity to study an *in*
*situ* bloom of UCYN-C. Complementary to this approach, (Pfreundt et al., Submitted-b) used
16S tag sequencing to examine heterotrophic bacterial diversity and successions during the
experiment and whether they evolved concurrently to that of diazotrophic and non-
diazotrophic phytoplankton groups. (Pfreundt et al., Submitted-a) used metatranscriptomics to
investigate the microbial gene expression dynamics from diazotrophic and non-diazotrophic
taxa and highlighted specific patterns of expression of genes involved in N, DIP, iron and
light utilization along the different phases of the experiment. (Leblanc et al., Submitted)
focused on the phytoplankton assemblages and dynamics along the experiment from pigment
signatures, flow cytometry and taxonomy analyses. In parallel, (Tedetti et al., 2015) used bio-
optical techniques to describe the spectral characteristics and the variability of dissolved and



particulate chromophoric materials according to the phytoplankton community composition
along the experiment. (Berman-Frank et al., Submitted) analyzed the spatial and temporal
dynamics of transparent exopolymeric particles (TEP), which are sticky carbon rich
compounds that are formed, degraded, and utilized in both biotic and abiotic processes, and
evaluated their role as an energy source for the auto- and heterotrophic communities.
Second, the bloom of diazotrophs (UCYN-C) obtained in the closed water body of the
mesocosms thanks to the DIP fertilization offered the opportunity to track the fate of DDN in
the ecosystem: (Berthelot et al., 2015) described the evolution of C, N, P pools and fluxes
along the experiment and investigated the contribution of $N_2$ fixation and DON use to primary
production and particle export. They also explored the fate of the freshly produced particulate
organic N, i.e. whether it was preferentially accumulated and recycled in the water column or
exported out of the system. Complementary to this approach (Knapp et al., Submitted) report
the results of a $\delta^{15}N$ budget performed in the manipulative mesocosms to assess the dominant
source of N (from $NO_3^-$ and/or $N_2$ fixation) fueling export production along the 23-days
experiment, and discuss how the measured geochemical signals correspond to concurrent
shifts in diazotroph and phytoplankton community composition. (Bonnet et al., Submitted)
explored the fate of DDN at shorter time scales during the height of the UCYN-C bloom and
investigated the relative contribution of each diazotroph phylotype to direct C export. They
also quantified the DDN released in the dissolved pool and its subsequent transfer to different
groups of plankton (picoplankton, diatoms) by using nanoSIMS coupled with $^{15}N_2$ isotopic
labelling. The same approach was used by (Berthelot et al., Submitted) to compare the DDN
transfer efficiency into non-diazotrophic plankton, whether it comes from *Trichodesmium*,
UCYN-C or UCYN-B. In parallel, (Hunt et al., Submitted) estimated the contribution of DDN
to zooplankton biomass in the mesoscosms based on naturel $^{15}N$ isotope values measurements
on zooplankton. They also studied the transfer of $^{15}N_2$ labelled phytoplankton to zooplankton
under contrasting situations (UCYN versus *Trichodesmium* versus Diatom-Diazotrophs
associations (DDAs) dominance), results that were complemented by qPCR assays on several
diazotroph phylotypes in zooplankton guts. (Spungin et al., Submitted) took advantage of the
*Trichodesmium* bloom occuring outside the mesocoms to specifically investigate its decline
and understand changes in genetic underpinning and features that could elucidate varying
stressors or causes of mortality of *Trichodesmium* in the natural environment.
Third, modelling was used at every stage of the project. Simulations performed with the
Eco3M-MED model have been used prior to the VAHINE experiment to help in the scientific
implementation of the project (timing and quantification of the DIP fertilization). (Gimenez et



al., Submitted) validated the model using the *in situ* data measured during the whole experiment, and provided additional information such as stoichiometry of planktonic organisms that could not be inferred through *in situ* measurements and offered the opportunity to deconvolute the different interlinked processes to help understanding the fate of DDN in oligotrophic ecosystems and its impact on carbon export.

Finally, a synthesis study by (Bonnet et al., In prep.) attempted to reconcile the diverse and complementary valuable methodological approaches used in this study to answer the scientific questions of the VAHINE project. After putting in perspective the different findings, the modelling approach has also been used here to investigate the impact of $N_2$ fixation on marine productivity, export and food web composition by artificially removing $N_2$ fixation in the model.

## Acknowledgements

Funding for this research was provided by the Agence Nationale de la Recherche (ANR starting grant VAHINE ANR-13-JS06-0002), the INSU-LEFE-CYBER program, GOPS and IRD. The authors thank the captain and crew of the R/V *Alis* as well as Riccardo-Rodolpho Metalpa for help in setting-up the moorings and Christophe Menkes for providing the surface chlorophyll map.

**Author contribution**: S. B. designed the experiments helped by T.M. J.M.G., F.L. designed the mesocosms, J.M.G., E.F., B.B., A.R. and J.M.B. deployed the mesocosms and performed CTD and traps sampling, M.R. analyzed CTD data, T.M was responsible for the nutrient analyses. S. Bonnet prepared the manuscript with contributions from all co-authors.

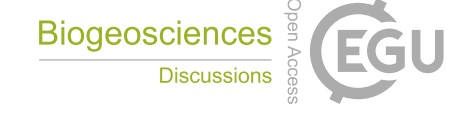

**Figure legends.**
**Figure 1.** Drawing representing the main features of the large-volume mesocosm device.
**Figure 2.** View of the experiment from the side and the seafloor during (a-c) and after the
deployment (d). e-f collect of sediment traps by the SCUBA divers (Photos: J.M. Boré and E.
Folcher, IRD).
**Figure 3.** Location of the study site of the VAHINE experiment. Map showing surface
chlorophyll a concentrations (MODIS) in the Southwestern Pacific during the study period
(January-February 2013), b) Map of the Noumea lagoon, c) a view taken from the Amédée
Island showing the location of mesocosms and R/V Alis.
**Figure 4.** Horizontal and vertical distributions of seawater temperature (°C), salinity and
fluorescence ($\mu$g L$^{-1}$) in M1 (a,e,i), M2 (b,f,j), M3 (c,g,k), and lagoon waters  (d,h,l). The grey
bars indicate the timing of the DIP spike on day 4.
**Figure 5.** Horizontal and vertical distributions of NO$_x$ and DIP ($\mu$mol L$^{-1}$) in M1 (a,e), M2
(b,f), M3 (c,g), and lagoon waters (d,h). The grey bars indicate the timing of the DIP spike on
day 4.













Figure 1.

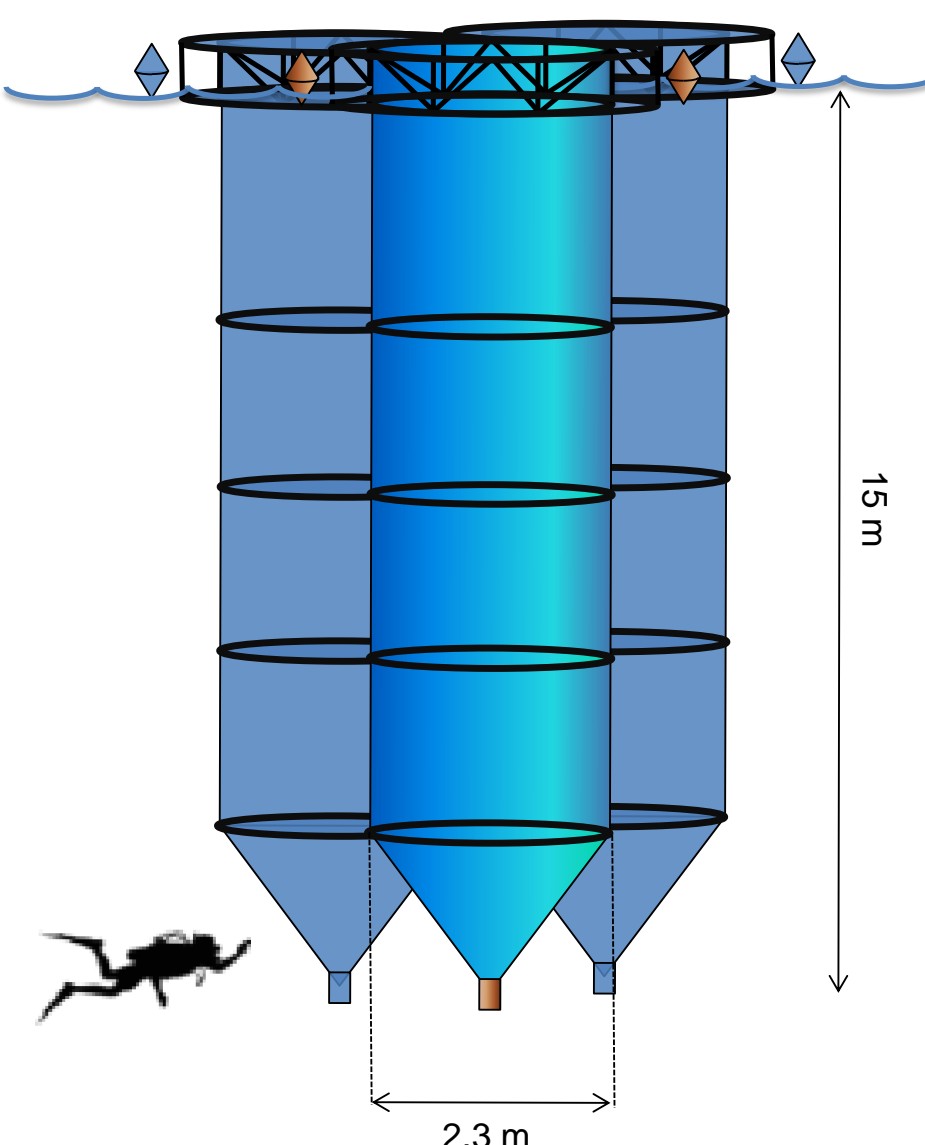



Figure 2.

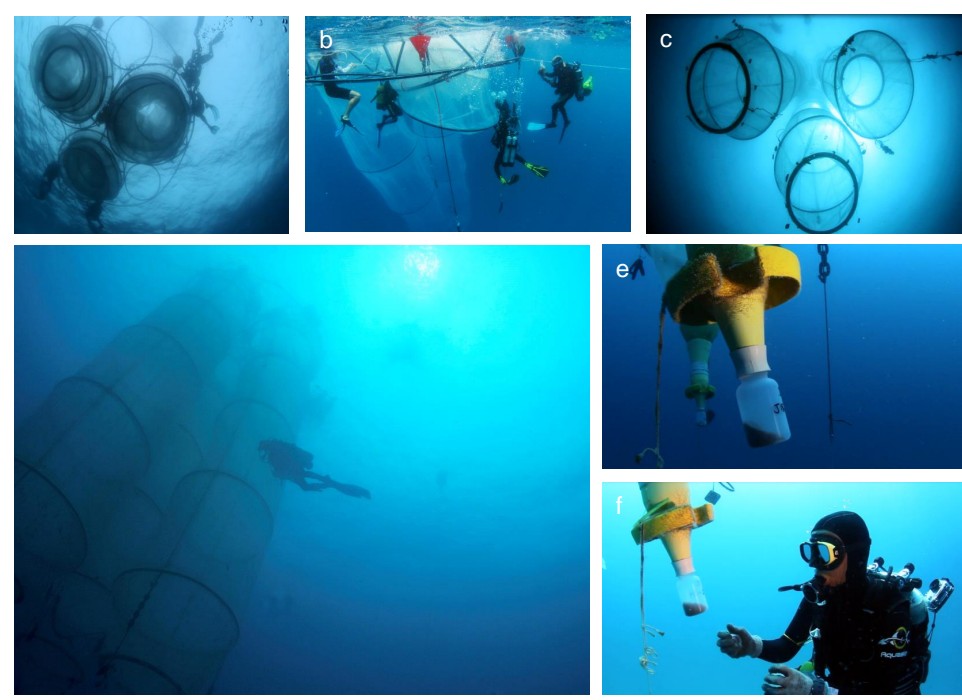




Figure 3.

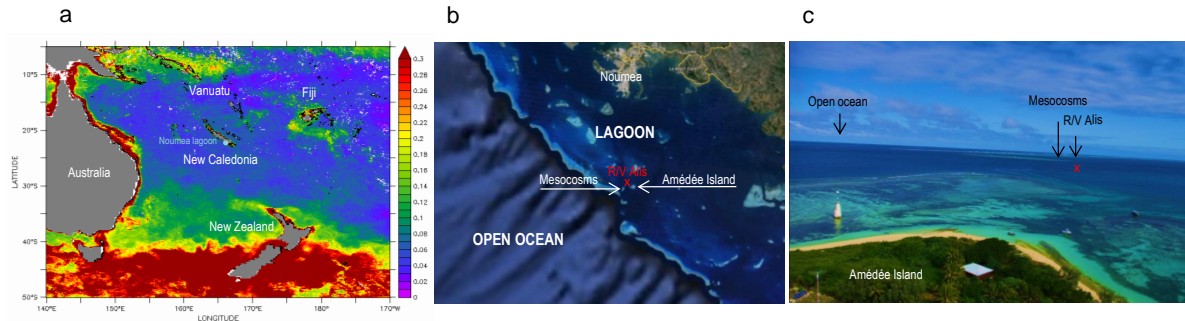



Figure 4.

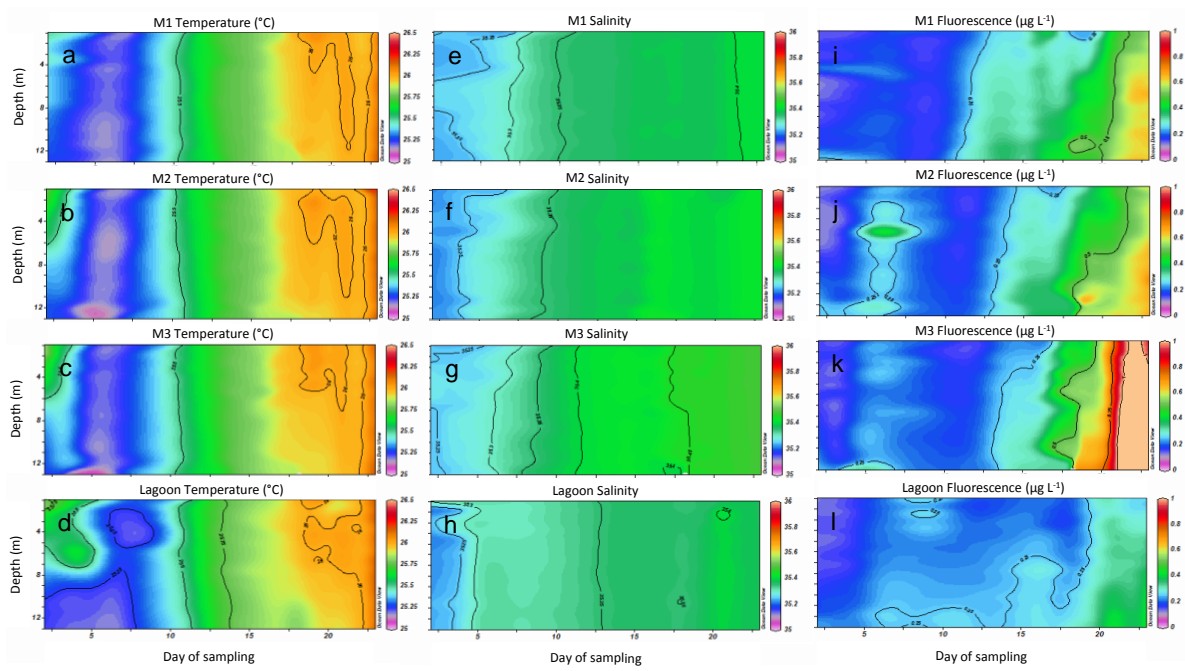





Figure 5.





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

| | Parameter measured | CV (%) between the three mesocosms |
|---|---|---|
| | $NO_3^-$ concentrations | 42 |
| | DON concentrations | 9 |
| | DOP concentrations | 21 |
| *Standing stocks* | PON concentrations | 21 |
| | POP concentrations | 26 |
| | Chl *a* concentrations | 26 |
| | TOC concentrations | 4 |
| | TEP concentrations | 24 |
| | Primary production | 29 |
| *Fluxes* | Bacterial production | 26 |
| | $N_2$ fixation | 34 |
| | *Prochlorococcus* abundances | 30 |
| | *Synechococcus* abundances | 30 |
| *Plankton abundances* | Pico-eukaryote abundances | 31 |
| | HNA abundances | 22 |
| | LNA abundances | 11 |
| | Average | 24 |



1  **Table 2.** Initial conditions (hydrological and biogeochemical parameters) recorded at 6 m-

2  depth just before the mesocosm deployment (January 13th).

| Temperature (°C) | Salinity | $[NO_3^-]$ ($\mu$mol L$^{-1}$) | [DIP] ($\mu$mol L$^{-1}$) | [Chl $a$ fluo] ($\mu$g L$^{-1}$) | [DON] ($\mu$mol L$^{-1}$) | [DOP] ($\mu$mol L$^{-1}$) | $N_2$ fixation (nmol N L$^{-1}$ d$^{-1}$) |
|---|---|---|---|---|---|---|---|
| 25.30 | 35.15 | 0.04±0.01 | 0.04±0.01 | 0.11 | 4.65±0.46 | 0.10±0.02 | 8.70±1.70 |

26