# Peer review of "Introduction to the project VAHINE: VAriability of vertical and tropHIc transfer of diazotroph derived N in the south wEst Pacific"

_Biogeosciences, 2015_

## Referee Comment (RC1) · J. Corredor (Referee) · 28 Jan 2016

This manuscript is a straightforward description of a mesocosm experiment denominated "VAriability of vertical and tropHIc transfer of diazotroph derived N in the south wEst Pacific" undertaken in waters of New Caledonia. The article is intended to serve as an introduction to a special volume on the subject by describing the goals of the experiment, the experimental setup, and the environmental conditions prevailing throughout the experiment's course. While the article is generally well written and structured, several grammatical errors and instances of improper use of the English language (e.g. past perfect instead of simple past) detract from its quality. I have taken the liberty of going through the manuscript using "track changes" to provide suggested rewrites of

these uncomfortable passages.

Two further comments:

- After what appears to have taken substantial exercise of the imagination, the authors came up with the acronym "VAHINE" choosing the appropriate lettering from the title (see above). It is puzzling then that the authors have not seen fit to explain, justify or even acknowledge this rather unusual acronym which, to my limited understanding of the Polynesian language, means "woman". Perhaps a sentence to that effect might be in order.

- While it is understandable that collaborating authors will want to set forth their results in detail in their individual articles, the reader of this manuscript is left largely in the dark as to results of the experiment: the fate the N fixed through P stimulation. Quoting from the manuscript:

The main scientific questions of the VAHINE project were:

i) What is the primary route of transfer of DDN through the planktonic food web, i.e. is DDN preferably transferred to large size (e.g. diatoms), small size (pico-, nanophytoplankton) phytoplankton, or to the microbial food web? How much DDN is transferred to zooplankton?

ii) Does the development of diazotrophs influence auto- and heterotrophic plankton diversity and gene expression dynamics, as well as pico-, nano-, and microphytoplankton abundances? Do they influence zooplankton dynamics?

iii) Does the development of diazotrophs significantly modify the stocks, fluxes or, ratios of the major biogenic elements (C, N, P)?

iv) Does the development of diazotrophs influences the efficiency of carbon export? Is this export direct or indirect?

Once again, a brief sentence or two describing the major findings appears to be in

order.

Jorge E. Corredor
* * *

---

## Referee Comment (RC2) · Anonymous Referee #2 · 1 Feb 2016

This is very interesting project that captures the readers attention from both an eco-logical and experimental perspective. The use of mesocosms in ecological studies has a long history in some areas of research but have been barely used in other settings. Their use for open-water studies is difficult and I can only imagine the logistics involved in setting up this month long study. Sophie Bonnet should be commended for leading the mesocosm project that appears to have been successful in increasing the abundance of diazotrophs through the addition of phosphorus.

A few comments are made below designed to increase the usefulness of this introductory paper.

Page 3 line 32-34, Page 4, line 1-7 The four stated research questions could be better

formulated. I would recommend removing the three subquestions or making them independent. Have you considered structuring them to indicate any priority in research? At the moment it sounds like you wanted to measure everything. You might also want to consider including a single sentence next to each of the research questions highlighting the results from the project to provide the reader with an immediate answer as to what was found.

Page 5 Line 7. I have not read the DUNE project work, but I recommend including some additional references to the use of mesocosms in ecological studies.

Page 16 Line 18 A black and white map would be equally useful as the color images shown demarking the position of the mesoscom experiment. I realize its bland but can be effective.

Page 7 Line 21 How was the concentration of DIP decided and what would have happened if you added more or less? Did you consider also adding iron or something micro-nutrient?

Page 8 Line 30 Why did you stop sampling after 23 days when you had the maximum fluorescence? I realize this probably relates to logistics, but clearly there were still changes occurring inside the bags that would have been good to capture

Figures: I am not sure that Figure 5a-c is necessary. I am not a big fan of the ocean data view color palette. For example it is difficult to interpret changes in fluorescence.

---

## Author Response (AR1)

Noumea, February 8th 2016; Marseille, February 11th 2016

Revisions [Manuscript bg-2015-615]

We thank both reviewers for very useful suggestions. We have addressed all of their concerns below, and in a revised manuscript. A point by point response to reviewers is below (comments are in italics with our replies below), and all new text in the corresponding manuscript is track change mode.

**Reviewer 1.**

*While the article is generally well written and structured, several grammatical errors and instances of improper use of the English language (e.g. past perfect instead of simple past) detract from its quality. I have taken the liberty of going through the manuscript using "track changes" to provide suggested rewrites of these uncomfortable passages.*
We are very grateful to the reviewer for providing such suggestions. They all have been taken into account in the revised version of the manuscript.

*After what appears to have taken substantial exercise of the imagination, the authors came up with the acronym "VAHINE" choosing the appropriate lettering from the title (see above). It is puzzling then that the authors have not seen fit to explain, justify or even acknowledge this rather unusual acronym which, to my limited understanding of the Polynesian language, means "woman". Perhaps a sentence to that effect might be in order.*
We agree that the acronym VAHINE is quite unusual. We have added a sentence page 3 line 30 to explain this choice: 'The acronym VAHINE (VAriability of vertical and tropHIc transfer of diazotroph derived N in the south wEst Pacific) was chosen in order to take reference to the Pacific culture where this experiment has been performed with the help of local people'. The project is also leaded by a woman and some readers might see a fully assumed feminist Act.

*While it is understandable that collaborating authors will want to set forth their results in detail in their individual articles, the reader of this manuscript is left largely in the dark as to results of the experiment: the fate the N fixed through P stimulation. Once again, a brief sentence or two describing the major findings appears to be in order.*
We decided not to provide one or two sentences of the major results after each scientific question, as it would require many explanations. We rather extended the section 4 of the manuscript 'Special issue presentation' and summarized the major results of each contributing paper to the special issue to provide the reader the main findings of this study. Section 4 has thus been totally modified.

**Reviewer 2.**

*Page 3 line 32-34, Page 4, line 1-7 The four stated research questions could be better formulated. I would recommend removing the three subquestions or making them independent. Have you considered structuring them to indicate any priority in research? At the moment it sounds like you wanted to measure everything.*

The three subquestions have been removed and the questions formulated as research priorities as follows: 'The main scientific research priorities of the project were:

i)      To quantify the DDN which enters the planktonic food web,

ii)      To investigate how the development of diazotrophs influences the subsequent diversity, gene expression, and production of primary producers, heterotrophic bacterioplankton, and subsequently the zooplankton abundance,

iii)      To examine whether different functional types of diazotrophs significantly modify the stocks and fluxes of the major biogenic elements (C, N, P),

iv)      To elucidate whether the efficiency of particulate matter export depends on the development of different functional types of diazotrophs.

*You might also want to consider including a single sentence next to each of the research questions highlighting the results from the project to provide the reader with an immediate answer as to what was found.*

Please see response to Reviewer 1: We decided not to provide one or two sentences of the major results after each scientific question, as it would require many explanations. We rather extended the section 4 of the manuscript 'Special issue presentation' and summarized the major results of each contributing paper to the special issue to provide the reader the main findings of this study. Section 4 has thus been totally modified.

*Page 5 Line 7. I have not read the DUNE project work, but I recommend including some additional references to the use of mesocosms in ecological studies.*

An additional sentence and aadditional references have been provided, among which a review paper on mesocosms (Stewart et al., 2013) page 5 line 15: 'Mesocosms are now widely used in ecological studies (Riebesell et al., 2013; Stewart et al., 2013) and enable isolation of water masses of several cubic meters from physical dispersion for several weeks…'.

The Stewart et al. reference has also been added page 5 line 25: 'Among the different types of mesocosms available (Stewart et al., 2013), the model of mesocosms chosen for this study (surface 4.15 $m^2$, volume ~50 $m^3$, Fig. 1) are sea-going mesocosms entirely transportable that can be used under low to moderate wind/wave conditions (20-25 knots/2.5 wave height). They have been designed in the framework of the DUNE project (Guieu et al., 2010; Guieu et al., 2014) and consist in large transparent bags made…'

*Page 16 Line 18 A black and white map would be equally useful as the color images shown demarking the position of the mesoscom experiment. I realize its bland but can be effective.*

Figure 3b has been replaced by a black and white map.

*Page 7 Line 21 How was the concentration of DIP decided and what would have happened if you added more or less?*

We decided to add 0.8 μM of phosphate because such concentrations have already been measured in the New Caledonian lagoon and pre-experiment modelling studies and were shown to be able to stimulate $N_2$ fixation. We knew that phosphate availability was the ultimate control of nitrogen inputs by $N_2$ fixation there (Moutin et al., 2005, 2008). Finally, simulated (thanks to the ECO3M model, see Gimenez et al., 2016, This issue) and experimental responses obtained after a 0.8 μM phosphate enrichment are close, and simulated response without phosphate enrichment show a very different and low response of the plankton community inside the mesocosms (Gimenez et al., 2016). The addition of more phosphate might not change so much the short term (20 days) response of the system because the time of the response is mainly controlled by the lag time between change in growth rate at the cell level and change in growth rate at the population level (Gimenez et al., 2016). The

following sentence has been added to the text: 'Such concentrations have already been measured in the New Caledonian lagoon and were shown to be able to stimulate $N_2$ fixation. The amount of DIP added was also chosen based on the modelling work performed by Gimenez et al. (2016), confirming a clear stimulation of $N_2$ fixation by 0.8 µmol $L^{-1}$ DIP in our experimental systems, and an absence of stimulation without any DIP enrichment.

*Did you consider also adding iron or something micro-nutrient?*
New Caledonian soils are very rich in metals. A third of its surface (5500 $km^2$) is covered by soils originating from ultramafic rocks (peridotites and serpentinites) which have exceptionally high levels of metals such as Fe, Ni, Cr, Co, and Mn. Lagoon waters are thus rich is metals as well and Fe in particular is not limiting for phytoplankton growth. We thus decided not to supplement the mesoscosm with trace metals. The following sentence has been added page 8 line 31: 'New Caledonian soils are very rich in metals. A third of its surface (5500 $km^2$) is covered by soils originating from ultramafic rocks which have exceptionally high levels of metals such as Fe, Ni, Cr, Co, and Mn (Jaffré, 1980). Consequently, dissolved trace metals are particularly abundant in the Noumea lagoon (Migon et al., 2007). Iron concentrations measured during the Diapalis cruises around New Caledonia were higher than those reported in the sub-tropical North Pacific and the high iron inputs in this region are hypothesized to drive the South West Pacific towards a DIP depletion (Van Den Broeck et al., 2004). Metals were thus not supplemented to the mesocosms'.

*Page 8 Line 30 Why did you stop sampling after 23 days when you had the maximum fluorescence? I realize this probably relates to logistics, but clearly there were still changes occurring inside the bags that would have been good to capture.*
We totally agree with this comment, it would have been very interesting to sample after day 23 as we reached the maxima of fluorescence at that period. The R/V Alis was assisting the project and for logistical reasons (time ship allocated to the project), it was planned in advance that the experiment could only last for 23 days. The Eco3M model platform (companion paper from Gimenez et al., 2016) was used to simulate carbon export up to day 35 and shows that C export greatly increased after day 23.

*Figures: I am not sure that Figure 5a-c is necessary. I am not a big fan of the ocean data view color palette. For example it is difficult to interpret changes in fluorescence.*
Nitrate concentrations were close to quantification limits during the course of the experiment, indicating that other sources of nitrogen (among which $N_2$ fixation) were providing the major sources of nitrogen to the system. We believe it is important to keep the nitrate plots in Figure 5 to inform the reader of the nitrate deficiency in the system with respect to phosphate.

**Literature cited**

Gimenez, A., Baklouti, M., Bonnet, S., and Moutin, T.: Biogeochemical fluxes and fate of diazotroph derived nitrogen in the food web after a phosphate enrichment: Modeling of the VAHINE mesocosms experiment, Biogeosciences Discussions, doi: doi:10.5194/bg-2015-611, 2016. 2016.

Guieu, C., Dulac, F., Desboeufs, K., Wagener, T., Pulido-Villena, E., Grisoni, J.-M., Louis, F., Ridame, C., Blain, S., Brunet, C., Bon Nguyen, E., Tran, S., Labiadh, M., and Dominici, J.-M.: Large clean mesocosms and simulated dust deposition: a new methodology to investigate responses of marine oligotrophic ecosystems to atmospheric inputs, Biogeosciences, 7, 2765-2784, 2010.

Guieu, C., Dulac, F., Ridame, C., and Pondaven, P.: Introduction to project DUNE, a DUst experiment in a low Nutrient, low chlorophyll Ecosystem, Biogeosciences, 11, 425-442, 2014.

Jaffré, T.: Etude écologique du Peuplement Végétal Des Sols Dérivés de Roches Ultrabasiques en Nouvelle-Calédonie, Paris, 1980.

Migon, C., Ouillon, S., Mari, X., and Nicolas, E.: Geochemical and hydrodynamic constraints on the distribution of trace metal concentrations in the lagoon of Noumea, New Caledonia, Estuarine, Coastal and Shelf Science, 74, 756-765, 2007.

Riebesell, U., Czerny, J., von Bröckel, K., Boxhammer, T., Büdenbender, J., Deckelnick, M., Fischer, M., Hoffmann, D., Krug, S. A., Lentz, U., Ludwig, A., Muche, R., and Schulz, K. G.: Technical Note: A mobile sea-going mesocosm system – new opportunities for ocean change research, Biogeosciences, 10, 1835-1847, 2013.

Stewart, R. I. A., Dossena, M., Bohan, D. A., Jeppesen, E., Kordas, R. L., Ledger, M. E., Meerhoff, M., Moss, B., Mulder, C., Shurin, J. B., Suttle, B., Thompson, R., Trimmer, M., and Woodward, G.: Mesocosm Experiments as a Tool for Ecological Climate-Change Research, Advances in Ecological Research, 48, 71-181, 2013.

Van Den Broeck, N., Moutin, T., Rodier, M., and Le Bouteille, A.: Seasonal variations of phosphate availability in the SW Pacific Ocean near New Caledonia, Marine and Ecological Progress Series, 268, 1-12, 2004.

[revised manuscript text omitted]

Important progress on the magnitude and the ecological role of marine $N_2$ fixation in biogeochemical cycles has been made by the international oceanographic community over the last two decades. They include the landmark discovery of unicellular diazotrophic organisms of pico- and nanoplanktonic size termed UCYN, e.g. (Zehr et al., 2001), and new and unexpected ecological niches where diazotrophs are active, such as N-rich oxygen minimum zones, e.g. (Dekaezemacker et al., 2013; Fernandez et al., 2011). Thus, we have gained a much better understanding of this process. However, a critical question that remains poorly studied is the fate of N newly fixed by diazotrophs (or diazotroph derived N, hereafter referred to as DDN) in oceanic food webs, and its impact on $CO_2$ uptake and export (BCP) (Mulholland, 2007). The VAHINE project proposes a scientific contribution to answer these questions, based on a combination of experimentation and modelling involving recently developed innovative techniques. The acronym VAHINE (VAriability of vertical and tropHIc transfer of diazotroph derived N in the south wEst Pacific) was chosen in order to take reference to the Pacific culture where this experiment has been performed with the help of local people. The main scientific research priorities of the  project were:

i)      To quantify the DDN which enters the planktonic food web,

ii)     To investigate how the development of diazotrophs influences the subsequent diversity, gene expression, and production of primary producers, heterotrophic bacterioplankton, and subsequently zooplankton abundance,

iii)    To examine whether different functional types of diazotrophs significantly modify the stocks and fluxes of the major biogenic elements (C, N, P),

iv)     To elucidate whether the efficiency of particulate matter export depends on the development of different functional types of diazotrophs.

i) What is the primary route of transfer of DDN through the planktonic food web, i.e. is DDN preferably transferred to large size (e.g. diatoms), small size (pico-, nanophytoplankton) phytoplankton, or to the microbial food web? How much DDN is transferred to zooplankton? ii) Does the development of diazotrophs influence auto- and heterotrophic plankton diversity and gene expression dynamics, as well as pico-, nano-, and microphytoplankton abundances? Do they influence zooplankton dynamics? iii) Does the development of diazotrophs significantly modify the stocks, fluxes, ratios of the major biogenic elements (C, N, P)? iv) Does the development of diazotrophs influences the efficiency of carbon export? Is this export direct or indirect?

Summarized conclusions of each article composing the special issue are provided in section 4 of this manuscript (Special issue presentation). Additionally, Aa detailed literature review on our knowledge regarding the fate of DDN in the ocean is provided in the synthesis article of the present issue (Bonnet et al., Submitted) together with a detailed description of the experimental and modelling results obtained during the project that answer the above scientific questions.

Below, Here we will focus on the technical challenges and the methods developed to answer the scientific questions of the project. .

[revised manuscript text omitted]

New Caledonian soils are very rich in metals. A third of its surface (5500 km$^2$) is covered by soils originating from ultramafic rocks which have exceptionally high levels of metals such as Fe, Ni, Cr, Co, and Mn (Jaffré, 1980). Consequently, dissolved trace metals are particularly abundant in the Noumea lagoon (Migon et al., 2007). Iron concentrations measured during the Diapalis cruises around New Caledonia were higher than those reported in the sub-tropical North Pacific and the high iron inputs in this region are hypothesized to drive the South West Pacific towards a DIP depletion. Metals were thus not supplemented to the mesocosms.

**2.4 Logistics, sampling strategy**

As the mesocosms were moored 28 km off the coast, all the experimental work had to be performed on site: scientific laboratories were setup on the R/V Alis (28.5 m) moored 0.5 nautical mile from the mesocosms, and on the Amédée sand island located one nautical mile from the mesocosms (Fig. 3b, c), on which we set up a laboratory and accommodated scientists for the duration of the VAHINE experiment.

Sampling in the mesocosms started on January 15[th] (day 2). The experiment lasted for 23 days for logistical reasons (i.e. It was performed daily for 23 days until February 6[th]) and sampling was performed daily at 7 am from the sampling platform moored next to the mesocosms. Every day after collection, seawater samples were immediately carried out to the R/V Alis and the Amédée for immediate processing.

Discrete samples were collected at three selected depths (1 m, 6 m, 12 m) in each mesocosm and outside (hereafter termed 'lagoon waters') using a braided PVC tubing connected to the Teflon PFA pump activated by pressurized air from diving tanks, allowing to sampling of large volumes with the least possible perturbation inside the mesocosms. For stocks measurements, 50-L PE carboys were filled at each depth of each mesocosm, immediately transported onboard the R/V Alis for subsampling and samples treatments. For fluxes measurements (primary production, bacterial production, N$_2$ fixation), samples were directly collected in incubation bottles and transported onboard to skip avoid the subsampling step and minimize the time between collection, tracer spikes and incubation. For prokaryotic diversity and gene expression measurements, 10-L carboys were filled (from M1 only) and carried out to the Amédée laboratory for immediate processing. A total of 220 L were sampled every day from each mesocosms, corresponding to ~10 % of the total mesocosms volume sampled at the end of the 23-days experiment.

After seawater sampling, vertical CTD profiles were performed (around 10 am) using a SBE 19 plus Seabird CTD in each mesocosm and outside the mesocosms to  document the vertical structure of temperature, salinity and fluorescence. The CTD *in situ* fluorescence data were fitted to the Chl *a* data from fluorometry measurements using a linear least squares regression.

Sediment traps were then collected daily from each mesocosm by two SCUBA divers (Fig. 2e, f1). They followed the same protocol everyday: they  gently tapped the cone of the mesocosms to dislodge  sinking material  retained on the walls, waited for 15 minutes, and collected the 250 mL flasks screwed to the trap system of each mesocosm and immediately replaced it  with a new one.

Vertical net hauls were performed every four days using a 30 cm diameter, 100 cm long, 80 µm mesh net fitted with a filtering cod end. On each sampling occasion, three vertical hauls were collected from each mesocosm and lagoon waters, representing a total volume of 2.13 $m^3$, i.e. 4 % of the total mesocosm volume. This sampling strategy was chosen to minimize the effect of zooplankton catches on the plankton abundance and composition in the mesocosms.

**2.5 Replicability among the mesocosms**

Guieu et al. (2010) and Guieu et al. (2014) have performed several mesocosm experiments in the Mediterranean Sea and demonstrated that the type of mesocosms used in the present study is well adapted to conduct replicated process studies on the first levels of the pelagic food web in LNLC environments. In order to evaluate the reproducibility among the three  mesocosms deployed during VAHINE, we calculated the coefficient of variation (CV, %) of the main stocks and fluxes measured every day for 23 days for every sampling depth (Table 1, the methods are described in detail in the publications composing this special issue). The CV ranged from 4 to 42 % depending on the parameter considered. It was  lowest for TOC and DON concentrations (4 and 9 %, respectively), which is very satisfying as these CV are close to the precision of the methods themselves, indicating a good reproducibility between mesocosms. It was  highest for $NO_3^-$ concentrations (42 %), which is consistent with the fact that $NO_3^-$ concentrations were close to quantification limits of conventional methods (~0.05 µmol $L^{-1}$) during the 23-days experiment: when the mean value is close to zero, the CV approaches infinity and is therefore sensitive to small changes in the mean. For flux measurements of PP, BP and $N_2$ fixation, the CV's were 29, 26 and 34%, respectively, which is also satisfying given the natural spatial heterogeneity of plankton

in the environment due to aggregation, (Seebah et al., 2014), or to the buoyancy of some diazotrophs such as *Trichodesmium* (Capone et al., 1997), which introduces  spatial variability, well known in the natural environment for $N_2$ fixation (Bombar et al., 2015). Another criterion to evaluate the consistency between mesocosms is to compare the evolution of the biogeochemical conditions and the plankton community composition between mesocosms. This approach  is described in details in several articles of the present issue and only some general features will be given here. As an example, bulk $N_2$ fixation rates averaged $18.5\pm1.1$ nmol N $L^{-1}$ $d^{-1}$ (standard deviation was calculated on the average $N_2$ fixation rates of each mesocosm) over the 23 days of the experiment  (all depths averaged together).  $N_2$ fixation rates did not differ significantly  among 
[revised manuscript text omitted]

dominated by Diatom-Diazotroph Associations (DDAs) during the first period of the experiment after the DIP fertilization (days 5 to 14; hereafter called P1), and a bloom of UCYN-C occurred during the second half (days 15 to 23, hereafter called P2), providing thean unique opportunity to compare the DDN transfer and export efficiency associated with different diazotrophs. This study provided the first growth rates for the uncultivated UCYN-A2 and the UCYN-C phylotypes, and the first opportunity to study an *in situ* bloom of UCYN-C. Complementary to this approach, Pfreundt et al. (2015) used 16S tag sequencing to examine the temporal dynamics of the prokaryotic community and observed clear successions of prokaryotes during the experiment, in relation with biogeochemical parameters. to examine heterotrophic bacterial diversity and successions during the experiment and whether they evolved concurrently to that of diazotrophic and non-diazotrophic phytoplankton groups. In a second study, Pfreundt et al. (Submitted) also used metatranscriptomics to investigate the microbial gene expression dynamics from diazotrophic and non-diazotrophic taxa and highlighted specific patterns of expression of genes involved in N, DIP, iron and light utilization along the different phases of the experiment. (Van Wambeke et al., (2015) revealed that heterotrophic bacterioplankton production and alkaline phosphatase activity were statistically higher during P2. Their results suggest that most of the DDN reached the heterotrophic bacterial community through indirect processes, like mortality, lysis and grazing. In parallel, Leblanc et al. (2016) focused on the phytoplankton assemblages and dynamics along the experiment from pigment signatures, flow cytometry and taxonomy analyses and revealed a monospecific bloom of the diatom *Cylindrotheca closterium* and an 2-fold increase in *Synechococcus* and nano-phytoeukaryotes during P2, concomitant with the UCYN-C bloom. In parallel, {Van Wambeke, 2015 #922}

Tedetti et al. (2015) used bio-optical techniques to describe the spectral characteristics and the variability of dissolved and particulate chromophoric materials according to the phytoplankton community composition and revealed a coupling between the dynamics of the $N_2$ fixation and that of chromophoric material in the South West Pacific through *Synechococcus* bloom. along the experiment. Berman-Frank et al. (2016) analyzed the spatial and temporal dynamics of transparent exopolymeric particles (TEP), which are sticky carbon rich compounds that are formed, degraded, and utilized in both biotic and abiotic processes, and evaluated measured a relatively stable TEP pool available as both a carbon source for plankton communities and facilitating aggregation and flux throughout the experimenttheir role as an energy source for the auto- and heterotrophic communities.

Second, the bloom of diazotrophs (UCYN-C) obtained in the closed water body of the mesocosms following DIP fertilization offered the opportunity to track the fate of DDN in the ecosystem: Berthelot et al. (2015) described the evolution of C, N, P pools and fluxes  during the course of the experiment and report a 3-fold increase in Chl *a* concentrations and $N_2$ fixation rates and a 5-fold increase in C export during the second half of the experiment (UCYN-C bloom). They also reveal that the *e*-ratio that quantifies the efficiency of a system to export particulate organic C  was significantly higher ($p < 0.05$) during P2 than during P1, indicating that the production sustained by UCYN-C was more efficient at promoting C export than the production sustained by DDAs.  Complementary to this approach Knapp et al. (2015) report the results of $\delta^{15}N$ measurements on DON, PON and particles from sediment traps and further substantiated these results with a significantly ($p<0.05$) higher contribution of $N_2$ fixation to export production during P2 ($56\pm24$ % and up to 80 % at the end of the experiment) compared to P1 ($47\pm6$ %). Bonnet et al. (2015) explored the fate of DDN at shorter time scales  and revealed that ~ 10 % of UCYN-C from the water column were exported daily to the traps, representing as much as $22.4 \pm 5.5$ % of the total POC exported at the height of the UCYN-C bloom. This export was mainly due to the aggregation of small ($5.7\pm0.8$ µm) UCYN-C cells into large (100–500 µm) aggregates. They also showed using a nanoSIMS approach that $21\pm4$ % of the DDN was transferred to non-diazotrophic plankton, mainly picoplankton ($18 \pm 4$ %) followed by diatoms ($3 \pm 2$ %) during P2.  The same nanoSIMS approach was used by Berthelot et al. (2016) in a parallel experimental study to compare the DDN transfer efficiency into non-diazotrophic plankton, whether it comes from UCYN-C, UCYN-B or *Trichodesmium*. They showed that the transfer was twice as high during a *Trichodesmium* bloom than during a UCYN-B or UCYN-C bloom, arguing that

filamentous diazotrophs blooms are more efficient at promoting non-diazotrophic production in N depleted areas. In parallel, Hunt et al. (2016) estimated a mean ~ 30 % contribution of DDN to zooplankton biomass the contribution of DDN to zooplankton biomass in the mesoscosms in the mesocosms based on naturael $^{15}$N isotope values measurements on zooplankton. They also provided evidence for direct ingestion and assimilation of UCYN-C-derived N by the zooplanktonstudied the transfer of $^{15}$N$_2$ labelled phytoplankton to zooplankton under contrasting situations (UCYN versus *Trichodesmium* versus Diatom-Diazotrophs associations (DDAs) dominance), results that were complemented by qPCR assays on several diazotroph phylotypes in zooplankton guts. Spungin et al. (2016) took advantage of the *Trichodesmium* bloom occuring outside the mesoscoms to specifically investigate its decline and understand changes in genetic underpinning and features that could elucidate varying stressors or causes of mortality of *Trichodesmium* in the natural environment.

Third, modelling was used at every stage of the project. Simulations performed with the 1D-vertical biogeochemical mechanistic Eco3M-MED model Eco3M MED model have been used prior to the VAHINE experiment to help in the scientific implementation of the project (timing and quantification of the DIP fertilization). Gimenez et al. (2016) validated the model using the *in situ* data measured during the whole experiment, and provided additional information such as stoichiometry of planktonic organisms that could not be inferred fromthrough *in situ* measurements and offered the opportunity to deconvolute the different interlinked biogeochemical processes occurring in the ecosystem to help understanding the fate of DDN in oligotrophic ecosystems and the its impact of N$_2$ fixation on carbon export. Finally, a synthesis study by Bonnet et al. (Submitted) attemptsed to summarize our knowledge and the unresolved questions regarding the fate of DDN in the ocean, synthetize and link the major experimental and modelling results obtained during the project and described in the VAHINE Special issue. It reconciles the diverse and complementary valuable methodological approaches used in this study to answer the scientific questions of the VAHINE project. After putting in perspective the different experimental findings, the modelling approach has also been used in the synthesis article as a tool here to investigate the impact of N$_2$ fixation on marine productivity, export and food web composition by artificially removing N$_2$ fixation in the model.

1 **Acknowledgements**

2 Funding for this research was provided by the Agence Nationale de la Recherche (ANR
3 starting grant VAHINE ANR-13-JS06-0002), the INSU-LEFE-CYBER program, GOPS and
4 IRD. The authors thank the captain and crew of the R/V *Alis* as well as Riccardo-Rodolpho
5 Metalpa for help in setting-up the moorings and Christophe Menkes for providing the surface
6 chlorophyll map.

8 **Author contribution**: S. B. designed the experiments helped by T.M. J.M.G., F.L. designed
9 the mesocosms, J.M.G., E.F., B.B., A.R. and J.M.B. deployed the mesocosms and performed
10 CTD and traps sampling, M.R. analyzed CTD data, T.M was responsible for the nutrient
11 analyses. S. Bonnet prepared the manuscript with contributions from all co-authors.

28

29

30

31

32

33

34

35

36

37

**Figure legends.**

**Figure 1.** Drawing representing the main features of the large-volume mesocosm device.

**Figure 2.** View of the experiment from the side and the seafloor during (a-c) and after the deployment (d). e-f collect of sediment traps by the SCUBA divers (Photos: J.M. Boré and E. Folcher, IRD).

**Figure 3.** Location of the study site of the VAHINE experiment. Map showing surface chlorophyll a concentrations (MODIS) in the Southwestern Pacific during the study period (January-February 2013), b) Map of the Noumea lagoon, c) a view taken from the Amédée Island showing the location of mesocosms and R/V Alis.

**Figure 4.** Horizontal and vertical distributions of seawater temperature (°C), salinity and fluorescence ($\mu g\,L^{-1}$) in M1 (a,e,i), M2 (b,f,j), M3 (c,g,k), and lagoon waters (d,h,l). The grey bars indicate the timing of the DIP spike on day 4.

**Figure 5.** Horizontal and vertical distributions of $NO_{3\,x}^{-}$ and DIP ($\mu mol\,L^{-1}$) in M1 (a,e), M2 (b,f), M3 (c,g), and lagoon waters (d,h). The grey bars indicate the timing of the DIP spike on day 4.

**Table 1.** Mean variation coefficients (CV = standard deviation x 100 / mean, %) calculated

for samples collected at the same time and the same depth in the three mesocosms. The CV

derived from these calculations was averaged over the 23-days experiment.

| | Parameter measured | CV (%) between the three mesocosms |
|---|---|---|
| | $NO_3^-$ concentrations | 42 |
| | DON concentrations | 9 |
| | DOP concentrations | 21 |
| *Standing stocks* | PON concentrations | 21 |
| | POPconcentrations | 26 |
| | Chl *a* concentrations | 26 |
| | TOC concentrations | 4 |
| | TEP concentrations | 24 |
| | Primary production | 29 |
| *Fluxes* | Bacterial production | 26 |
| | $N_2$ fixation | 34 |
| | *Prochlorococcus* abundances | 30 |
| | *Synechococcus* abundances | 30 |
| *Plankton abundances* | Pico-eukaryote abundances | 31 |
| | HNA abundances | 22 |
| | LNA abundances | 11 |
| | Average | 24 |

1 **Table 2.** Initial conditions (hydrological and biogeochemical parameters) recorded at 6 m-

2 depth just before the mesocosm deployment (January 13th).

| Temperature (°C) | Salinity | [NO$_3$] (µmol L$^{-1}$) | [DIP] (µmol L$^{-1}$) | [Chl *a* fluo] (µg L$^{-1}$) | [DON] (µmol L$^{-1}$) | [DOP] (µmol L$^{-1}$) | N$_2$ fixation (nmol N L$^{-1}$ d$^{-1}$) |
|---|---|---|---|---|---|---|---|
| 25.30 | 35.15 | 0.04±0.01 | 0.04±0.01 | 0.11 | 4.65±0.46 | 0.10±0.02 | 8.70±1.70 |